# Laser Scanning Guided Localization Imaging with a Laser-Machined Two-Dimensional Flexible Ultrasonic Array

**DOI:** 10.3390/mi13050754

**Published:** 2022-05-10

**Authors:** Jianzhong Chen, Wei Liu, Dianbao Gu, Dawei Wu

**Affiliations:** 1State Key Laboratory of Mechanics and Control of Mechanical Structures, Nanjing University of Aeronautics and Astronautics, Nanjing 210016, China; jzc@nuaa.edu.cn (J.C.); liuw2288@nuaa.edu.cn (W.L.); 2Xinhua Hospital Chongming Branch, Shanghai 202150, China

**Keywords:** flexible ultrasound array, surface imaging, single-layer “island bridge”, array design

## Abstract

Advances in flexible integrated circuit technology and piezoelectric materials allow high-quality stretchable piezoelectric transducers to be built in a form that is easy to integrate with the body’s soft, curved, and time-dynamic surfaces. The resulting capabilities create new opportunities for studying disease states, monitoring health/wellness, building human–machine interfaces, and performing other operations. However, more widespread application scenarios are placing new demands on the high flexibility and small size of the array. This paper provides a 8 × 8 two-dimensional flexible ultrasonic array (2D-FUA) based on laser micromachining; a novel single-layer “island bridge” structure was used to design flexible array and piezoelectric array elements to improve the imaging capability on complex surfaces. The mechanical and acoustoelectric properties of the array are characterized, and a novel laser scanning and positioning method is introduced to solve the problem of array element displacement after deformation of the 2D-FUA. Finally, a multi-modal localization imaging experiment was carried out on the multi-target steel pin on the plane and curved surface based on the Verasonics system. The results show that the laser scanning method has the ability to assist the rapid imaging of flexible arrays on surfaces with complex shapes, and that 2D-FUA has wide application potential in medical-assisted localization imaging.

## 1. Introduction

Ultrasound imaging is an essential adjunct to modern medicine and its technology is widely used to visualize the interior of objects for non-destructive evaluation, health monitoring, and medical treatment due to its non-invasive, high-accuracy, high-sensitivity, and strong penetration capabilities [1,2,3]. Ultrasound probes for conventional medical imaging applications are almost always rigid and bulky, with the transducer requiring external clamps to hold it in place and the patient requiring a fixed posture confined in a specially designed frame [4]. Flexible ultrasonic transducer is an art of combining flexible circuit design with rigid piezoelectric ceramic phase of ultrasound; rigid piezoelectric materials can be laminated in a way suitable for a variety of complex surfaces, opening a functional window for the realization of dynamic and static ultrasound diagnosis, ultrasound therapy, ultrasound imaging, and other medical aids [5,6]. Flexible transducers are capable of producing sound waves with ultra-high frequencies, which have irreplaceable advantages over conventional medical imaging such as portability, ease of access, and ease of design and operation [7,8].

Therefore, many studies have been devoted to the development of flexible ultrasound devices to achieve medical aids for imaging and treatment by means of extracorporeal apposition (attached skin, head, etc.) [9,10] and human–machine interface (guided puncture, localized debridement, etc.) [11,12], and are not limited to scenarios such as hospitals allowing more widespread use of the modality. Flexible ultrasonic array fabrication has strict requirements for electrical connections, tensile properties, material design, and other techniques, and the operating characteristics are the result of each array element being excited by electrical circuit control [13]. Often encouraged by the growing needs for high diagnostic/therapeutic efficacy and for new fields of applications, the development of advanced array has been an active research topic with ever-challenging and ambitious technical requirements. Flexible array design and fabrication is more interested in size reduction, increased sensitivity, reduced number of elements, and wide bandwidth [14,15,16]. Ultrasonic array imaging places higher demands on increased flexibility, reduced array elements, and algorithm matching [17,18,19]. Flexible ultrasound array working on flat surfaces can achieve similar acoustic imaging functions to rigid ultrasound probes. The development of a portable, human-fitted flexible ultrasound device for ultrasound imaging of animals and even humans, dynamic health detection, and even neuromodulation of the human brain is expected to be based on flexible ultrasound technology [20].

However, when a flexible ultrasound array is attached to a complex curved surface, the relative positions of the array elements change depending on the shape of the surface and can no longer be considered to be equally spaced. It is necessary to redefine the relative positions between the array elements when the flexible array is attached to various curved surfaces. There are several methods to solve the non-planar state array localization problem by collecting information on the relative positions of the array elements attached to the target surface by computed tomography (CT) [21,22] or by obtaining the time of flight between each array element and the target point [23,24]. The time reversal method is often selected to solve the array element coordinate offset, which is realized in three ways [25,26]. The first is that the micro acoustic reflector is placed at the target point to recapture the reflected sound wave by each array element in the initial array. The second is to implant the point sound source into the target point so that each array element on the initial array can detect its spherical wave. The last way is to directly implant the micro pressure sensor into the position of the target point to detect the sound waves independently emitted by each array element on the initial array, and directly obtain the time of flight (TOF) data with the target point. However, the main problem with these methods is that they are invasive, and the relative position of array elements is obtained indirectly through TOF. Therefore, a non-invasive laser scanning technology is used to directly obtain the offset array element position, which can avoid invasive implantation, and explore a non-invasive, low-cost, and fast array element positioning method in curved surface state.

This paper presents an 8 × 8 flexible ultrasonic array with a single-layer “island bridge” structure that can be used for imaging curved surfaces. Firstly, a single-layer “island bridge” is introduced to optimize the array structure and circuit. The array element size is simulated to determine the optimal size. The mechanical and acoustoelectric properties of the 2D-FUA are characterized. Secondly, a laser scanning positioning method is proposed to solve the problem of array element displacement after flexible array deformation, and the surface and experimental coordinates are fitted through displacement transformation. This method is non-invasive space-time scanning. Finally, the multimodal imaging and positioning experiment of multi-target steel needle is carried out based on a Verasonics system. The results show that laser scanning can assist the flexible array in the rapid imaging of complex shapes, and the flexible two-dimensional ultrasound array has broad potential application prospects in medical assisted positioning imaging.

## 2. Array Design and Surface Guided Positioning

### 2.1. Piezoelectric Array Element Design

The most effective vibration mode of the ultrasonic transducer is the longitudinal vibration mode of the piezoelectric arrays, and too much transverse vibration between the arrays will weaken the longitudinal waves entering the medium. It is necessary to optimize the size of the piezoelectric material (PZT-8 thickness 0.5 mm) to ensure that the piezoelectric array elements can be in longitudinal vibration mode in the design. PZFlex finite element simulation software simulates and designs the piezoelectric array element without setting the backing and matching layer. Figure 1 shows the simulation results of electrical impedance and phase angle of piezoelectric array elements in different sizes. Piezoelectric array elements with different side lengths show an obvious main peak of resonance frequency, and the resonance frequency will shift to low frequency with the increase of piezoelectric array element side length. It should be noted that when the side length of the piezoelectric array element is greater than 1 mm, the main peak of resonance frequency is relatively obvious, and there is no secondary peak interference of other obvious vibration modes. The working mode of the piezoelectric array is relatively single. When the side length of a piezoelectric array element is less than 0.9 mm, the secondary peak of resonance frequency is particularly obvious, and the main peak of resonance frequency will be affected by the secondary peak, indicating that the longitudinal vibration of the piezoelectric array element will be disturbed by other vibration modes, and the emission efficiency will be reduced. Therefore, the minimum side length of 0.9 mm is selected as the final size of the piezoelectric array element.

### 2.2. Flexible Electrode Design

As shown in Figure 2, there are 64 controllable array elements in 2D-FUA which makes it difficult for traditional electrode connection to achieve effective control. Stretchable flexible electrode is designed based on the row column addressing principle. The double-sided copper film covered by PI film undertakes the circuit interconnection, and 16 row and column electrodes are provided on each side of the array to connect N2 inactive elements. The electrode and circuit design were completed before packaging with silica gel material. We strip 8 “row” copper films on the top of the array to obtain a single control column matrix of leads, and we strip 8 “column” copper films in the vertical direction on the bottom of the array. The non-stripped copper film electrode forms row and column circuit addressing in the vertical direction, which can activate the corresponding row circuit and column circuit with external independent leads as shown in Figure 2. Row and column addressing electrode configuration has the ability to reduce the number of lead elements from N^2^ to N + N. As shown in Figure 2a, the activated array element area is jointly determined by the selected row electrode and column electrode. The principle that the excitation signal directly activates the corresponding independent array element or the activated array element area passes through the external lead is shown in Figure 2b. The design of N + N flexible electrodes allows the array to realize independent excitation and orderly excitation of any unit with 16 leads, which helps to reconstruct the shape of targets in multi-section images.

### 2.3. Design of Single-Layer “Island Bridge” Array

Single-layer “island bridge” structure based on laser micromachining was used to design 2D-FUA. The array elements are arranged in a matrix of 8 × 8 array electrodes, and the circuit and flexible interconnection are realized between each unit in a single-layer “island bridge” structure. As shown in Figure 3a, polyimide film (PI film 0.2 mm) is wrapped and pretreated by conductive copper film. A single-layer “island bridge” is processed directly on the surface of PI film by laser. The square groove provides a solid and reliable framework for rigid PZT (PZT-8). The single-layer “island bridge” has simple structure and is easy to manufacture. Compared with the “island-bridge” structure proposed by Xu et al., the single-layer “island-bridge” structure can exhibit higher flexibility to adapt to complex surfaces [27,28]. Figure 3b shows the structure and function of a single-layer “island” in the 2D-FUA. The silver coating wraps the top and bottom of the PI film. The silver paste coating on both sides of the array element provides excellent electrical interconnection, effectively suppresses the ringing effect (excessive vibration), and improves the axial resolution of the pattern. An improved superior serpentine hinge is applied to realize array “island” and circuit interconnection [29]. The optimized superior serpentine hinge is composed of a basic half hinge and an antisymmetric half hinge, and the right angle connection is used at both ends of the hinge to replace the corner connection, which can better resist the repeated shear stress at the hinge connection in tension. The thickness of the entire flexible hinge matrix is designed to be 0.25 mm, the width of the serpentine hinge is 0.2 mm, the center distance between the serpentine hinges is 0.4 mm, and the spacing of the array elements is 3.4 mm.

### 2.4. Laser Scanning Guided Surface Array Element Positioning

The laser scanning guided array element positioning flow chart is shown in Figure 4. The focus and angle of the laser scanning are first adjusted, and the exposure and gain of the camera are adjusted, and then the calibration points are identified by scanning through a special calibration plate, which is used to determine the scan space before the formal scanning. In the next step, a multi-angle spatiotemporal scan of the curved glassware with the 2D-FUA is performed to obtain the complete scanned model. Finally, the scanned model is imported into Geomagic Studio software for post-processing, such as noise reduction, surface trimming, stitching, and merging.

Laser scanning is an inverse modeling technique that can quickly and non-invasively acquire the surface shape of the target object and capture the relative positions of the array elements attached to the surface. Figure 5a shows the laser scanning system, which consists of a laser scanner (Hangzhou Jusen Technology Co., Ltd., Hangzhou China), a target object, and a computer. The 2D-FUA is attached to the outer surface of the curved glassware as the scanning target, and black and white circular markers are attached to the array and the glassware as the scanning identification points. These identification points allow for the stitching together of different angle scanned models in the software to obtain a complete surface profile.

The brown sphere is the identification point of the scan, which reconstructs the surface shape of the glassware and clearly shows the position of each element. The scanned model was imported into Geomagic Studio software for noise reduction, surface trimming, stitching, and merging, and was converted into a surface file in .igs format for transferring graphics files between different software. The scanned model after post-processing is shown in Figure 5b. The scanned model was imported into Creo Parametric software, and marker points were created at the center of each array element, and the 3D absolute coordinates of the marker points were obtained directly using the software measurement tools.

The scanning parameters are listed in Table 1. As shown in Figure 5b, the glassware was sprayed with an inverse enhancement coating and the scanned identification points automatically reconstructed the surface shape of the glassware in order to clearly see the positions of the individual array elements. In order to fit the coordinates in the scanning model with those of the ultrasonic imaging system, it is necessary to establish a transformed coordinate system to realize the conversion of global, local, and relative coordinates. The origin of the global coordinate system is defined as absolute zero, and the *x*-axis direction is the same as the x′ direction of the local coordinate system, omitting the transformation coordinates. Therefore, only the y and z axes of the global coordinate system need to be converted to plane coordinates. A marker point is established at the center of each array element to convert the absolute coordinates of the array element to Verasonics coordinates. The global coordinate system z-y is converted and then rotated 53° counterclockwise to obtain the transition coordinate system z″-y″.

## 3. Results and Discussion

### 3.1. Mechanical Properties Characterization

As shown in Figure 6, the superior serpentine hinge is able to recover its initial state at the ultimate stretch length relying on the hinge rebound force. The array still works positively under 40–60% simultaneous biaxial stretching conditions, and the array maintains excellent conductivity at the extreme stretching state. PI films and copper films may develop fatigue cracks or even fail during plastic deformation, and the mechanical properties of the array are also expressed in terms of the fatigue resistance of the device, which must be able to maintain mechanical integrity during repeated loading. As shown in Figure 6a–c, the 2D-FUA is verified in practice by stretching, twisting, and bending to verify the mechanical properties of the array and the encapsulator. The low modulus (low modulus −70 kpa) and tensile properties of the silicon thin film material, as shown in Figure 6d,e, provide a compatible platform for the device to seal various shapes of components in the array. The silicon material that seals the array provides a thickness of less than 15 mm silicon elastic film for the liner and bottom layers, which not only provides insulation and adhesion between adjacent layers, but also provides acoustics and device robustness to effectively avoid acoustic second resonance. Polyimide square groove and silicon-filled material can effectively avoid transverse vibration, reduce crosstalk, and induce longitudinal waves into the target body. As shown in Figure 6f, the encapsulation material (silicone) can rely on its own van der Waals forces to adsorb on the surface of the body, and the flexibility can be stretched to adapt to various curved surfaces and bumps.

### 3.2. Acoustic Performance Characterization

The impedance analyzer (Agilent 4294A) was used to measure and characterize the electrical impedance and acoustic properties of the 2D-FUA. As shown in Figure 7b, the resonant frequency, fr, and anti-resonant frequency, fa, of the array were 1.95 MHz and 2.19 MHz, respectively, and the phase angle of the transducer was about −13.2°. The acoustic performance of the array was evaluated in water with an Olympus 5072PR pulse receiver. As shown in Figure 7a, the 2D-FUA was attached to the inner wall of the water tank, and the flexible array was excited at resonant frequencies for pulse-echo experiments. Figure 7c shows the pulse–echo response of the array to verify that the array has excellent transmitting and receiving acoustic performance. Crosstalk between 8 × 8 flexible ultrasonic arrays of piezoelectric elements was simulated by PZFlex (now renamed OnScale). The simulation results show that the maximum crosstalk between the cells of the array is −32 dB, which satisfies the theoretical requirement that the maximum crosstalk value should be less than −35 dB [30,31].

### 3.3. Curved Surface Imaging Experiment

As shown in Figure 8, the 2D-FUA curved surface imaging platform is built based on the Verasonics system. The 2D-FUA is affixed along the curved surface of the inner wall of the water tank, and the array performs B-type ultrasound scanning based on the array element spatial coordinates on the randomly placed steel column target to obtain the location of the target object by imaging. As shown in Figure 8b, the curved glass tank was filled with deionized water (acoustic impedance of water is about 1.5 MRayl, acoustic impedance of biological soft tissue is about 1.6 MRayl), and the electrode wire of the 2D-FUA was connected to the adapter of the ultrasound platform panel interface through an adapter plate. The glassware was 100 mm in diameter and the wire was 1 mm in diameter without matching and backing layers. Since the scanning area of the 2D-FUA is a spatial cube, a B-scan of the surface array can yield any 2D cross section. The dimension of the cross section in the x-direction is defined as the width of the array without bending, and the depth in the z-direction is defined as 12. All the array elements transmit ultrasonic waves simultaneously and reconstruct the received signals on the two-dimensional cross section at once. In addition to this, planar experiments of 2D-FUA were also provided to further validate the flexible array imaging performance in Figure 8a.

### 3.4. Experimental Results and Discussion

Figure 9a shows the results of the 2D-FUA planar scan imaging with three randomly placed steel pins in the sink at the theoretical distances of 52 mm, 55 mm, and 70 mm from the array plane. The bright spots represent the three randomly placed steel pin positions, and the bright spots occupy the vertical axis coordinate values representing the measured wavelength distance between the steel pins and the array plane. Three different bright spot locations were captured, and the wavelength–distance position relationship was calculated to obtain 52.8 mm for the ellipse center of point a#, 56.8 mm for point b#, and 72.9 mm for point c#. The results show that the target distance measurements are very close to the theoretical values, with accuracy errors of 1.54%, 3.27%, and 4.14%, respectively.

Figure 9b shows the imaging results of the 2D-FUA, which verify that the laser scanning technique identifies the position of the array element, and the discrete state can assist in imaging. The positions of the targets can be identified by the white bright spots in the coordinate system, and the vertical axis values of the white bright spots directly correspond to the wavelengths of the three targets 1#, 2#, and 3# in relation to the array. By locating the white bright spot, the longitudinal coordinate of the elliptical center point of 1# bright spot is 53.8 mm, the longitudinal coordinate of 2# bright spot is 54.4 mm, and the longitudinal coordinate of 3# bright spot is 58.3 mm. The measured values are very close to the theoretical values of 52 mm, 53 mm, and 56 mm, respectively. The above experimental results verify the proposed laser scanning guided 2D-FUA imaging method.

Figure 9 shows that the imaging results have artifacts using both imaging methods, and this phenomenon is more obvious on the curved surface. Figure 9a shows that the bright spot 1# artifact is elliptical, which indicates that there is an accuracy error between the measurement result and the actual size of the wire (diameter 1 mm). The errors were mainly due to the fact that the 2D-FUA was not set up with a backing, which resulted in some acoustic energy not being transmitted, and that there was no matching layer between the array encapsulation material and the target.The imaging resolution can be improved by using 1–3 piezoelectric composites with low acoustic impedance and further optimizing the structure, such as reducing the array pitch. The work verifies that the designed 2D-FUA can be used for localization imaging, which provides a novel idea for medical flexible imaging. The localization imaging method using laser scanning guidance proposed in this paper can realize the function of contactless, spatiotemporal scanning and high-precision positioning, which has the ability to help the flexible array imaging positioning and solve the difficult problem of array element displacement during surface imaging.

## Figures and Tables

**Figure 1 micromachines-13-00754-f001:**
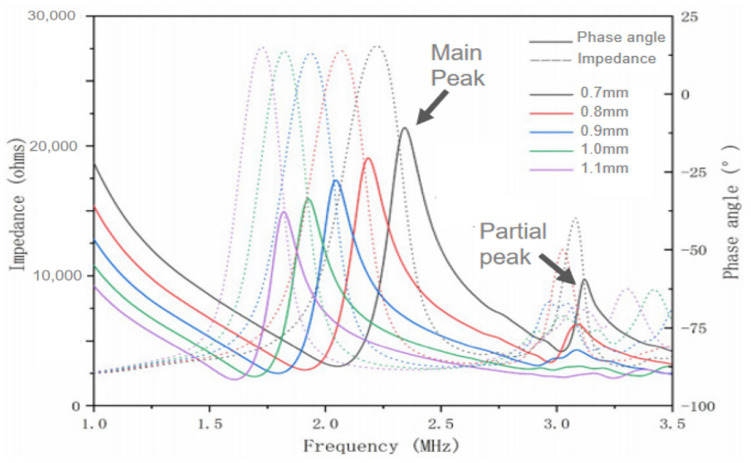
Simulation results of electrical impedance and phase angle.

**Figure 2 micromachines-13-00754-f002:**
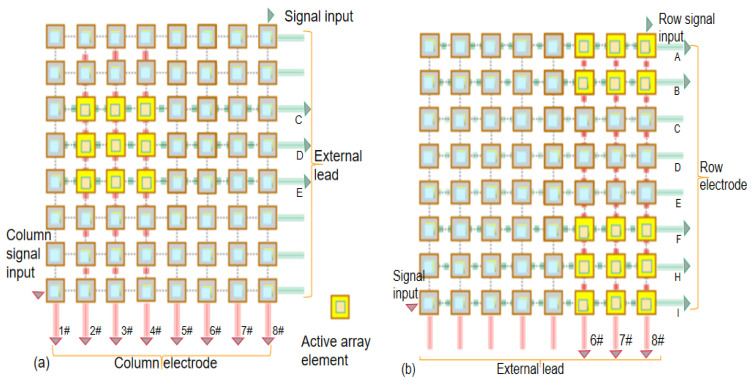
Flexible electrode activation: (**a**) column electrode, (**b**) row electrode.

**Figure 3 micromachines-13-00754-f003:**
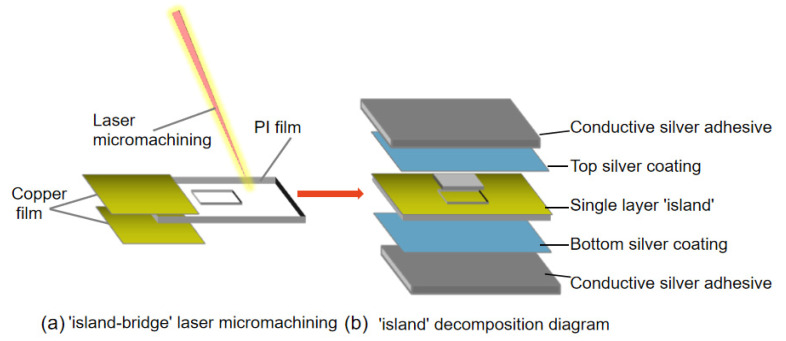
Single-layer “island bridge” structure.

**Figure 4 micromachines-13-00754-f004:**
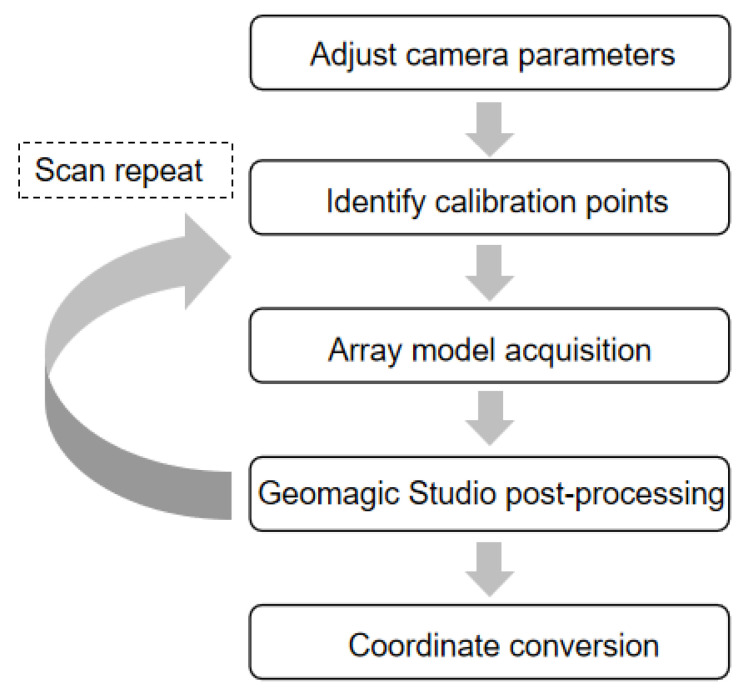
Laser scanning flow chart.

**Figure 5 micromachines-13-00754-f005:**
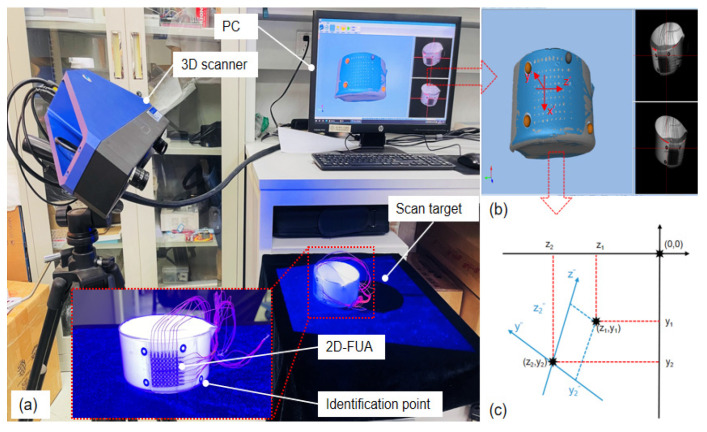
(**a**) Laser scanning; (**b**) scan morphology; (**c**) coordinate conversion in Verasonics.

**Figure 6 micromachines-13-00754-f006:**
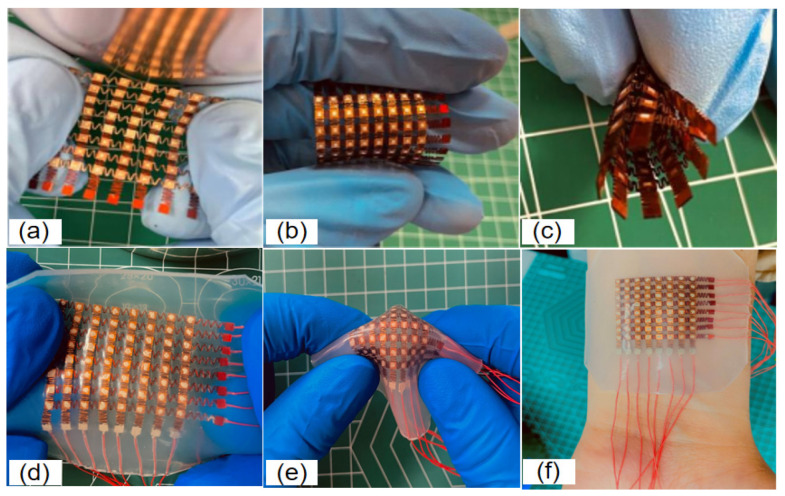
(**a**–**c**) Electrode flexible display; (**d**–**f**) array flexible display.

**Figure 7 micromachines-13-00754-f007:**
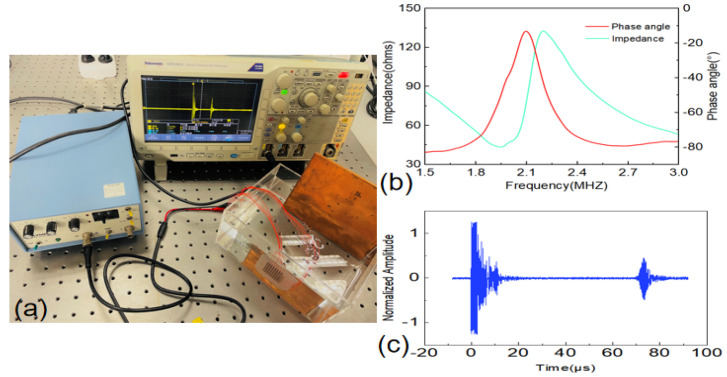
(**a**) Impulse echo; (**b**) impedance and phase angle spectra; (**c**) impulse echo response.

**Figure 8 micromachines-13-00754-f008:**
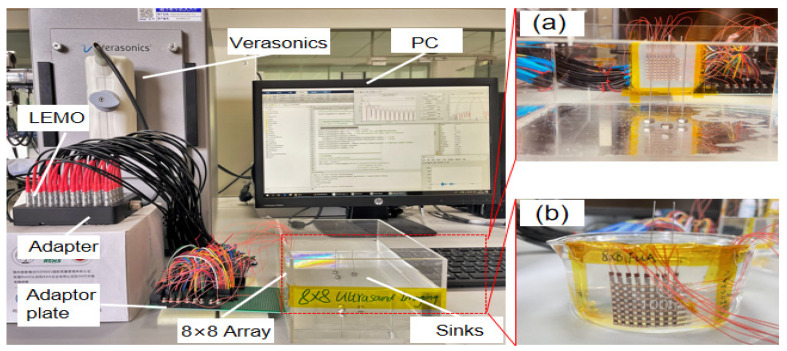
2D-FUA imaging platform: (**a**) planar; (**b**) curved surface.

**Figure 9 micromachines-13-00754-f009:**
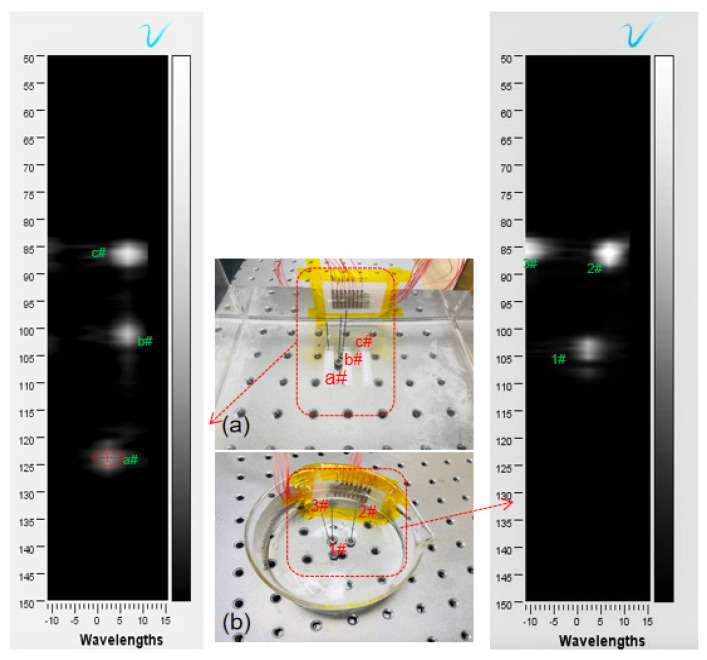
(**a**) Planar imaging; (**b**) curved surface imaging.

**Table 1 micromachines-13-00754-t001:** Basic parameters of the laser scanning.

Parameter	Specification
Light source	Blue light
Scanning mode	Binocular scanning
Scanning method	Raster scan
Scanning precision	50 μm
Single scanning time	0~5 s
Scan range	30 × 30 × 30 cm

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
