# Peer review of "Laser Scanning Guided Localization Imaging with a Laser-Machined Two-Dimensional Flexible Ultrasonic Array"

_micromachines, 2022, doi:10.3390/mi13050754_

Round 1

Reviewer 1 Report

Although the stretchable ultrasonic transducer can adapt complex surface,variation in distance between elements makes imaging difficult. Therefore, spatial localization of different elements is very important for stretchable ultrasonic transducer. This study presents an effective and reliable method to localize the elements. Before publication, there are several issues should be improved.

Abstract “a novel single-layer "island  bridge" structure was used to design flexible array and piezoelectric array elements to improve the  imaging capability on complex surfaces ”.  What is the different between your structure and Xu’s structure (Ref 29,30).

“It is necessary to 8×8 the cutting size of PZT-8 (thick- 97 ness 0.5 mm) in the 2D-FUA is designed ”.  Please rewrite the sentence.

“so that the piezoelectric array element can be in an economical and effective longitudinal vibration mode”. What the economical vibration mode is.

It can adapt to more complex curved surfaces and show more flexibility. Com- pared with the multi-layer 'island bridge' electrode proposed by Xu et al. Please rewrite the sentence.

Figure 3a. The relationship between Pl, pzt8 and copper film is not clear.

“An improved Superior serpentine hinge is applied to realize array 'island' and circuit inter- connection.“ Schematic and photo of the superior serpentine hinge are necessary. Besides,figure‘s resolution in this manuscript is poor.

In discussion and conclusion section, authors should emphasize the advantages of the laser scanning and positioning method,and compare it with previous studies.

Author Response

Dear Reviewer

Thank you very much for your comments on our paper, which were valuable and helpful. We have read each of the comments and have done our best to make corrections and additions to the corresponding pair in the text. The details are as follows:

Comment 1:Abstract “a novel single-layer "island  bridge" structure was used to design flexible array and piezoelectric array elements to improve the  imaging capability on complex surfaces ”. What is the different between your structure and Xu’s structure (Ref 29,30).

Answer:Thank you for your kind and helpful review comments. As you mentioned, the single-layer 'island bridge' structure is an important part of the array design structure in this paper.As shown in the figure, the 'island-bridge' structure proposed by XU et al. is prepared by stacking multiple layers and the process is complicated.The single-layer 'island-bridge' structure proposed in this paper is in direct laser processing using polyimide film (wrapped with conductive copper film).The advantages lie in the following two aspects.The possibility of generating a variety of shapes using mapping software, which can then be quickly processed and prepared by laser micromachining systems.The single-layer 'island-bridge', with its simple structure and controllable thickness, can provide a compatible platform for a wide range of components.      

Figure: Structural processing (a) Multilayer island bridge structure proposed by XU et al. (b) Single-layer island bridge structure with direct laser processing in this paper

Figure (a) citing literature :Mohan, A. V., Kim, N., Gu, Y., Bandodkar, A. J., You, J. M., Kumar, R., ... & Wang, J. (2017). Merging of Thin‐and Thick‐Film Fabrication Technologies: Toward Soft Stretchable “Island–Bridge” Devices. Advanced Materials Technologies,2(4), 1600284.

Comment 2:“It is necessary to 8×8 the cutting size of PZT-8 (thick- 97 ness 0.5 mm) in the 2D-FUA is designed ”. Please rewrite the sentence.

Answer:Thank you for your comment, it was very helpful.The article needs a clearer theoretical description in the '2.1 Piezoelectric array element design' section.We have redescribed the array element design and simulation and the changes have been marked in red for you.

The most effective vibration mode of the ultrasonic transducer is the longitudinal vibration mode of the piezoelectric arrays, and too much transverse vibration between the arrays will weaken the longitudinal waves entering the medium.It is necessary to optimize the size of the piezoelectric material (PZT-8 thickness 0.5 mm) to ensure that the piezoelectric array elements can be in longitudinal vibration mode in the design.

Comment 3:“so that the piezoelectric array element can be in an economical and effective longitudinal vibration mode”. What the economical vibration mode is

Answer:Thank you for this valuable and helpful comment.A more efficient vibration mode refers to the ultrasonic transducer design that allows the array element to vibrate to produce more longitudinal waves. We have rewritten this section of the article to make it easier for the reader to understand. The modified part is marked in red font in the paper for you.

The most effective vibration mode of the ultrasonic transducer is the longitudinal vibration mode of the piezoelectric arrays, and too much transverse vibration between the arrays will weaken the longitudinal waves entering the medium.It is necessary to optimize the size of the piezoelectric material (PZT-8 thickness 0.5 mm) to ensure that the piezoelectric array elements can be in longitudinal vibration mode in the design.

Comment 4:It can adapt to more complex curved surfaces and show more flexibility. Com- pared with the multi-layer 'island bridge' electrode proposed by Xu et al. Please rewrite the sentence.

Answer:Thank you for your comment, we have rewritten the grammatical errors.In addition to this, the full writing was checked for grammar, and the corrections are marked in red.

Compared with the 'island-bridge' structure proposed by Xu et al, the single-layer 'island-bridge' structure can exhibit higher flexibility to adapt to complex surfaces. [29,30].

Comment 5:Figure 3a. The relationship between Pl, pzt8 and copper film is not clear.

Answer:We sincerely thank you for your comment.We have modified Figure 3 as shown in the figure.In order to make it easier for the reader to understand the relationship between pzt8 and 'Island', the title of Figure 3 (a) is changed from PI and copper film to: 'Island-bridge' laser micromachining, and the title of Figure 3 (b) is changed to 'Island' decomposition diagram.

Comment 6:“An improved Superior serpentine hinge is applied to realize array 'island' and circuit inter- connection.“ Schematic and photo of the superior serpentine hinge are necessary. Besides,figure‘s resolution in this manuscript is poor

Answer:We thank the reviewers for their meaningful review comments. The advanced serpentine hinge is a previous research work of the authors.As shown in the figure, we have optimized and simulated the serpentine hinge to reduce the tensile stress by changing the corner design.We have done our best to adjust the resolution of the images, and in addition, we have cited the advanced serpentine hinge in the article with attribution.

An improved Superior serpentine hinge is applied to realize array 'island' and circuit interconnection[31].

[31]Liu, W., Chen, W., Zhu, C., & Wu, D. (2021). Design and micromachining of a stretchable two-dimensional ultrasonic array. Micro and Nano Engineering,13, 100096.

Figure: Advanced serpentine hinge stretching comparison and simulation

Figure (a) citing literature :Liu, W., Chen, W., Zhu, C., & Wu, D. (2021). Design and micromachining of a stretchable two-dimensional ultrasonic array. Micro and Nano Engineering,13, 100096.

Comment 7:In discussion and conclusion section, authors should emphasize the advantages of the laser scanning and positioning method,and compare it with previous studies.

Answer:The review comments you provided were very valuable and helpful.In the discussion and conclusion section, we have added a section on the advantages of the laser scanning positioning method, and the added section is marked in red.

The work verifies that the designed 2D-FUA can be used for localization imaging, which provides a novel idea for medical flexible imaging.The localization imaging method using laser scanning guidance proposed in this paper can realize the function of contactless,spatio-temporal scanning and high precision positioning, which has the ability to help the flexible array imaging positioning and solve the difficult problem of array element displacement during surface imaging.

We deeply appreciate helpful and valuable comments on our manuscript. If you have any queries, please don’t hesitate to contact me at the address below.

Thank you and best regards.

Yours Sincerely

Prof. Wu

E-mail: dwu@nuaa.edu.cn

Reviewer 2 Report

Dr. Wu and his team demonstrated a flexible and stretchable ultrasonic array with excellent mechanical properties under different modes of deformations. The size of each element was designed under the guidance of simulation, and a row-column connection strategy was used to control the array. To determine the location of each element on a curved surface, a 3D laser scanner was used to reconstruct the target and device with updated coordinates of each element. The imaging result has been demonstrated with a phantom wire. Overall, this manuscript achieves the level of Micromachines and is suggested to be accepted. Some minor comments are listed below:

  1. Compared with individual control of each element, the row-column connection may induce more severe electrical cross-talk. Please characterize the cross-talk level and solve this problem if the value is over -30 dB.
  2. Some details should be mentioned in the manuscript, for example, the branch and model of the 3D laser scanner, and the pitch of the array.
  3. The transmitting method and imaging reconstruction method can be briefly introduced.
  4. Reference 30 does not mention anything about the multi-layer 'island bridge' electrode. Please correct it.

Author Response

Dear Reviewer

Thank you very much for your comments on our paper, which were valuable and helpful. We have read each of the comments and have done our best to make corrections and additions to the corresponding pair in the text. The details are as follows:

Comment 1: Compared with individual control of each element, the row-column connection may induce more severe electrical cross-talk. Please characterize the cross-talk level and solve this problem if the value is over -30 dB.

Answer:Thank you for your review comment, which were helpful and valuable on this study.The array electrical crosstalk is a key parameter affecting the ultrasound transducer, and we simulate the lateral crosstalk between the array elements in the paper, and the results show -32 dB.According to the ultrasonic transducer design principle, crosstalk values higher than -35 dB can seriously affect the imaging quality.‘The simulation results show that the maximum crosstalk between the cells of the array is -32 dB, which satisfies the theoretical requirement that the maximum crosstalk value should be less than -35 dB[32,33].

In order to reduce the lateral crosstalk effect, we chose a 200 µm polyimide film to reduce the crosstalk between the array elements, in addition to using silicone as an encapsulation material.As described in 'Experimental results and discussion', 1-3 piezoelectric composites are used in subsequent studies to reduce crosstalk and improve the imaging performance of the flexible array.

Comment 2: Some details should be mentioned in the manuscript, for example, the branch and model of the 3D laser scanner, and the pitch of the array.

Answer:Thanks to your valuable comment, we have added detailed information about the model and array parameters of the laser scanner. The additions are highlighted in red for you.

The thickness of the entire flexible hinge matrix is designed to be 0.25 mm, the width of the serpentine hinge is 0.2 mm, the center distance between the serpentine hinges is 0.4 mm, and the spacing of the array elements is 3.4 mm.

Fig. 5(a) shows the laser scanning system, which consists of a laser scanner(Hangzhou jusen Technology Co., Ltd.Model:Sparrw), a target object and a computer.

Comment 3: The transmitting method and imaging reconstruction method can be briefly introduced.

Answer:Thank you for your valuable comment, which will make the article more understandable to the readers. Transmission methods and image reconstruction methods are added in the section 'Laser scanning guided surface array element positioning'.

The brown sphere is the identification point of the scan, which reconstructs the surface shape of the glassware and clearly shows the position of each element.The scanned model was imported into Geomagic Studio software for noise reduction, surface trimming, stitching and merging, and was converted into a surface file in .igs format for transferring graphics files between different software.The scanned model after post-processing is shown in Fig.5(b).The scanned model was imported into Creo Parametric software, and marker points were created at the center of each array element, and the 3D absolute coordinates of the marker points were obtained directly using the software's measurement tools.

Comment 4:Reference 30 does not mention anything about the multi-layer 'island bridge' electrode. Please correct it.

Answer:Thank you very much for your comment, we have replaced the literature [30].,and the article of XU et al. on 'island-bridge' is also added in the cited literature section.

[30]Wang, C. , Wang, C. , Huang, Z. , & Xu, S. . (2018). Materials and structures toward soft electronics. Advanced Materials, 30(50), 1801368.1-1801368.49.

[30]Mohan, A. V., Kim, N., Gu, Y., Bandodkar, A. J., You, J. M., Kumar, R., ... & Wang, J. (2017). Merging of Thin‐and Thick‐Film Fabrication Technologies: Toward Soft Stretchable “Island–Bridge” Devices. Advanced Materials Technologies, 2(4), 1600284.

We deeply appreciate helpful and valuable comments on our manuscript. If you have any queries, please don’t hesitate to contact me at the address below.

Thank you and best regards.

Yours Sincerely

Prof. Wu

E-mail: dwu@nuaa.edu.cn

Reviewer 3 Report

Chena et al. reported a novel method to improve the accuracy of the ultrasonic tomographic imaging assisted by a laser positioning technology. The authors simulated the resonance of piezoelectric element in advance, designed flexible array of piezoelectric elements and electrodes, fabricated devices, and investigated the imaging performance using a known sample. It seems highly original research and development, especially the accurate positioning of piezoelectric elements by using a laser system deserve attention. At present stage, the manuscript needs some trivial improvement, but I think that this work is a worth publishing in micromachines.

My comments on the contents can be found below:

Comment 1: Laser scanning positioning is a non-invasive method, but it is not the only non-invasive method. Are there any other positioning methods? Three-dimensional information can be able to be obtained by a traditional stereo camera.

Comment 2: The authors mentioned “The measured values are very close to the theoretical values” in line 272. However, distances measured by ultrasonic echoes were finite errors. What is the prime factor to make errors?

Comment 3: How much error is allowed for practical use such as medical inspections? Do the experimental results of ultrasonic echo location meet authors’ goals?

Comment 4: There is no explanation for the effect of the accuracy of the position of piezoelectric elements on the measurement results. The authors’ claim of this development is to measure the position of the piezoelectric elements accurately, the effect should be quantitatively investigated and qualitatively explained.

Author Response

Dear Reviewer

Thank you very much for your comments on our paper, which were valuable and helpful. We have read each of the comments and have done our best to make corrections and additions to the corresponding pair in the text. The details are as follows:

Comment 1: Laser scanning positioning is a non-invasive method, but it is not the only non-invasive method. Are there any other positioning methods? Three-dimensional information can be able to be obtained by a traditional stereo camera.

Answer:Thank you for your valuable and helpful comment. Laser scanning is an accurate, non-invasive method of obtaining array element positions after array deformation.In this paper we use a laser scanner with two scanning lenses, which requires model reconstruction and data import after scanning.The purpose of this method is to reconstruct a 3D image from the scan and then input the array element position coordinates into the Verasonics imaging system.

There are several methods that can solve the problem of array element localization in the non-planar state, such as:collecting the relative position information of the array elements attached to the target surface by computed tomography (CT) or by obtaining the time of flight (Time of Flight , TOF) between each array element and the target point.However, the main problem of these methods is that they are invasive and the relative position of the array elements is obtained indirectly by TOF.

To solve this problem, further research work is to investigate non-invasive dynamic localization imaging using optical fiber attached to a line array or a surface array for dynamic localization.

Comment 2: The authors mentioned “The measured values are very close to the theoretical values” in line 272. However, distances measured by ultrasonic echoes were finite errors. What is the prime factor to make errors?

Answer:Thank you for your valuable and helpful comment.According to the experimental imaging results and data analysis, the flexible ultrasound array can achieve localization imaging within a certain error range, and the surface imaging error is found to be larger by comparing the results of planar imaging and surface imaging.The main reason for this error in localization imaging is that the flexible array is stretched and the array element spacing becomes larger resulting in imaging error.This problem can be compensated by using optimized array structures or 1-3 piezoelectric composites for gain compensation.

Comment 3: How much error is allowed for practical use such as medical inspections? Do the experimental results of ultrasonic echo location meet authors’ goals?

Answer:Thank you very much for your valuable comment. This paper initially explores a precise, non-invasive method of guiding localization imaging using laser scanning to determine the position of array elements after array deformation.Secondly, conventional medical ultrasound probes are mostly rigid and bulky. As shown in the figure, the purpose of this paper is to initially prepare a flexible, lightweight and adaptable curved surface ultrasound transducer for imaging and localization.The preliminary experimental results are in accordance with the research objectives, although not to the level of medical precision but provide a novel idea for flexible transducer medical imaging.

Figure: Ultrasonic transducer light weight and flexibility demonstration

Comment 4: There is no explanation for the effect of the accuracy of the position of piezoelectric elements on the measurement results. The authors’ claim of this development is to measure the position of the piezoelectric elements accurately, the effect should be quantitatively investigated and qualitatively explained.

Answer:Thank you very much for your valuable and helpful comment.The main purpose of this paper is to realize a lightweight and stretchable flexible array, and realize laser scanning guided array element positioning imaging.A non-invasive laser scanning method is preliminarily explored to determine the position of array elements.The scanning system realizes the information transmission and image reconstruction of the array element. The accuracy of the actual position and expected position of the array element is determined by the scanning system. The two positions are reflected in absolute coordinates and relative coordinates. The data content is shown in the table.Absolute coordinates(x,y,z);Relative coordinates(x’,y’,z’)

Further work can realize flexible array imaging on a determined surface with known array element position, and compare the array element position in the imaging coordinate system and quantitatively analyze the error.In addition, to facilitate the reader's understanding of the steps of this method, we have added the transmission method and image reconstruction contents in the paper, and the additions are marked in red.

The brown sphere is the identification point of the scan, which reconstructs the surface shape of the glassware and clearly shows the position of each element.The scanned model was imported into Geomagic Studio software for noise reduction, surface trimming, stitching and merging, and was converted into a surface file in .igs format for transferring graphics files between different software.The scanned model after post-processing is shown in Fig.5(b).The scanned model was imported into Creo Parametric software, and marker points were created at the center of each array element, and the 3D absolute coordinates of the marker points were obtained directly using the software's measurement tools.

Array element marker points

x

y

z

x'      y'

z'

PNT12

30.769

-21.454

-31.164

8.54

5.27

-0.56

PNT13

PNT14

PNT15

PNT16

PNT17

PNT18

PNT19

PNT20

PNT21

PNT22

PNT23

26.831

27.007

27.057

27.159

27.237

27.350

23.532

23.580

23.823

23.821

23.880

-8.036

- 11.056

- 13.870

- 16.575

- 19.087

-21.551

-8.265

- 11.177

- 14.044

- 16.701

- 19.317

-20.988

-22.699

-24.527

-26.556

-28.692

-31.080

-20.956

-22.622

-24.507

-26.511

-28.710

-8.30

-4.86

- 1.51

1.87

5.16

8.57

-8.13

-4.81

- 1.38

1.95

5.36

1.33

1.51

1.56

1.66

1.74

1.85

- 1.97

- 1.92

- 1.68

- 1.68

- 1.62

-0.61 - 1.06 - 1.30 - 1.30 - 1.11 -0.69 -0.78 - 1.20 - 1.42 - 1.42

- 1.23

PNT24

PNT25

PNT26

PNT27

PNT28

23.953

20.189

20.264

20.407

20.424

-21.700

-8.411

- 11.287

- 14.184

- 16.883

-31.062

-20.911

-22.590

-24.471

-26.497

8.68

-8.04

-4.74

- 1.29

2.08

- 1.55

-5.31

-5.24

-5.09

-5.08

-0.79 -0.90 - 1.29 - 1.53

- 1.54

PNT29

PNT30

PNT31

PNT32

PNT33

20.542

20.675

16.798

16.968

17.108

- 19.469

-21.804

-8.540

- 11.439

- 17.289

-28.693

-31.051

-20.825

-22.545

-24.422

5.47

8.75

-7.99

-4.64

1.16

-4.96

-4.83

-8.70

-8.53

-8.39

- 1.34 -0.86 - 1.05 - 1.42

-3.44

PNT34

PNT35

PNT36

17.111

17.212

17.233

- 16.982

- 19.548

-22.021

-26.406

-28.588

-31.017

2.11

5.47

8.91

-8.39

-8.29

-8.27

- 1.67 - 1.47

- 1.02

We deeply appreciate helpful and valuable comments on our manuscript. If you have any queries, please don’t hesitate to contact me at the address below.

Thank you and best regards.

Yours Sincerely

Prof. Wu

E-mail: dwu@nuaa.edu.cn
